# Dynamics of the microbiota in patients with *Clostridioides difficile*: Recurrence, treatment, sex, and immunosuppression

Maria Paz Ventero[1☉‡], Rocio Herrero[2☉‡], Iryna Tyshkovska[1],
Maria-Dolores Valverde-Fredet[2], Juan Carlos Rodríguez[1,3*],
Miguel Rodríguez-Fernández[2], Pilar González-De-La-Aleja[4], Marta Trigo[2], Monica Parra[1],
Ana-Isabel Aller[2], Silvia Otero[4], Reinaldo Espindola-Gomez[2], José Manuel Ramos[3,4],
Eva M. León[2], Maria García[5], Miguel Nicolas Navarrete-Lorite[6], Jara Llenas-García[3,5,7],
Ines Portillo[6], Francisco Jover[8], Maria Tasias[9], Juan Jose Caston[10], Concepción Gil[11],
David Vinuesa-Garcia[12], Cristina Gomez-Ayerbe[13], Francisco J. Martínez Marcos[14],
Nicolas Merchante[2,3‡], Esperanza Merino De Lucas[2,3‡]

1 Microbiology Service, Alicante General University Hospital - Alicante Institute of Health and Biomedical Research (ISABIAL), Alicante, Spain, 2 Clinical Unit of Infectious Diseases and Microbiology, Valme University Hospital, Institute of Biomedicine of Seville (IBiS), University of Seville, Sevilla, Spain, 3 Clinical Medicine Department, Miguel Hernández University, Elche, Spain, 4 Unit of Infectious Diseases, Alicante General University Hospital - Alicante Institute of Health and Biomedical Research (ISABIAL), Alicante, Spain, 5 Unit of Infectious Diseases, Vega Baja University Hospital - Orihuela, Spain-FISABIO, Foundation for the Promotion of Health and Biomedical Research of the Valencian Community, Valencia, Spain, 6 Unit of Infectious Diseases, Virgen Macarena University Hospital, Sevilla, Spain, 7 CIBERINFEC, Carlos III Health Institute, Madrid, Spain, 8 Unit of Infectious Diseases, Sant Joan D'Alacant University Hospital, Sant Joan D'Alacant, Spain, 9 Infectious Diseases Service, La Fe University and Polytechnic Hospital, Valencia, Spain, 10 Unit of Infectious Diseases, Reina Sofía University Hospital, Córdoba, Spain, 11 Unit of Infectious Diseases, Marina Baixa Hospital, VillaJoiosa, Spain, 12 Unit of Infectious Diseases, San Cecilio University Hospital, Granada, Spain, 13 Unit of Infectious Diseases, Virgen de la Victoria University Hospital, Malaga, Spain, 14 Unit of Infectious Diseases, Juan Ramón Jimenez University Hospital, Huelva, Spain

☉ These authors contributed equally to this work.
‡ MPV and RH share first authorship on this work. NM and EMDL are joint senior authors on this work.
* rodriguez_juadia@gva.es

## Abstract

### Background

Alterations in the gut microbiome are central to the pathogenesis and recurrence of *Clostridioides difficile* infection (CDI).

### Objective

To evaluate intestinal microbiome changes during CDI and their association with recurrence, sex, age, and immunosuppression.

### Methods

Patients from the CDI-ANCRAID-SEICV cohort were consecutively enrolled. Stool samples were obtained at diagnosis (Dx), end of treatment (ET), and eight weeks

**Data availability statement:** The sequencing raw data are available in NCBI, Bioproject PRJNA1313452.

**Funding:** This study was funded by respective grants to NM from the Institute of Health Carlos III (ISCIII) (PI20/01450) and to JMR from the Ministry of Education, Culture, Universities and Employment of the Valencian Community (CIAICO/2023/274) and to MPV received from the Alicante Institute for Health and Biomedical Research (ISABIAL) (project 2024/B/29). The MPV's salary was funded by a grant for hiring technical staff: the "ESF Investing in your future" along with a PTA Grant (PTA2021-020215-I) funded by MCIN/AEI/10.13039/501100011033. The funders had no role in study design, data collection and analysis, decision to publish, or preparation of the manuscript.

**Competing interests:** The authors have declared that no competing interests exist.

post-treatment (8W) or upon recurrence. Microbiota composition was analyzed by 16S rRNA sequencing using QIIME2 and R. Outcomes were compared by demographics, immunosuppression, and treatment (vancomycin [VNC], vancomycin-bezlotuzumab [VNC-BZL], fidaxomicin [FDX]).

## Results

Among 143 patients, non-recurrent cases showed higher biodiversity at 8W versus diagnosis (H $p = 0.002$, ASVs $p < 0.001$), unlike recurrent cases. Diversity decreased with VNC (H $p > 0.001$, ASVs $p < 0.001$) but was preserved with FDX (H $p = 0.15$). Recovery of Shannon diversity was limited in women ($p = 0.50$) and immunocompromised patients ($p = 0.31$). At ET, *Fusobacteria* and *Verrucomicrobiota* were less abundant in recurrent than non-recurrent cases (0.77%, 0.53% vs 3.43%, 3.50%). FDX-treated samples showed higher *Bacteroidetes* (31.33%) compared to VNC (5.23%) or VNC-BZL (3.12%). Women exhibited increased *Firmicutes* abundance ($p = 0.034$).

## Conclusions

Restoration of microbial diversity correlates with CDI resolution. FDX preserves gut microbiota better than VNC or VNC-BZL. Women and immunocompromised patients demonstrate impaired microbiota recovery.

## Author summary

*Clostridioides difficile* infection is a major cause of antibiotic-associated diarrhea and is known for its high risk of recurrence. Alterations in the gut microbiome, a complex community of microorganisms that plays a key role in intestinal health, have been associated with both the risk of recurrence and the clinical course of the disease. In this study, we followed patients with *C. difficile* infection over time to better understand how the gut microbiome changes during treatment and recovery, and how these changes relate to recurrence, treatment type, sex, age, and immune status. Stool samples were collected at diagnosis, at the end of treatment, and several weeks later, and analyzed using DNA sequencing to characterize the intestinal microbiota. Patients who did not experience recurrence showed a progressive recovery of microbial diversity, whereas this recovery was limited in patients with recurrent infection. Treatment choice also influenced microbiome preservation: fidaxomicin was associated with better maintenance of microbial diversity compared with vancomycin-based regimens. In addition, women and immunocompromised patients showed impaired microbiota recovery. These findings highlight the importance of microbiome restoration in the resolution of *C. difficile* infection and suggest that both patient characteristics and treatment strategies may influence clinical outcomes.

## Introduction

*Clostridioides difficile* infection (CDI) is a leading cause of healthcare-associated diarrhea and is associated with increased morbidity and mortality. One of the most significant clinical challenges in managing CDI is its high recurrence rate, affecting approximately 25% of patients following initial treatment. This high recurrence rate underscores the critical need to understand the factors that differentiate sustained clinical resolution from relapse [1,2]. A hallmark of CDI is the disruption of the gut microbiota, characterized by a loss of microbial diversity and alterations in community composition. This dysbiosis creates a favorable environment for the proliferation of *C. difficile*. Consequently, the restoration of a healthy and resilient gut microbiota is essential for achieving long-term resolution of the infection [3,4].

Numerous studies have generated evidence on the microbiome in patients with recurrent CDI in an effort to identify microbial signatures associated with susceptibility to relapse. However, relatively few have provided analogous data on the microbiome dynamics in patients who achieve sustained clinical cure. Comparative analyses of these two patient populations may provide critical insights into the protective role of the gut microbiota and help identify specific taxa or microbial patterns associated with resistance to recurrence [5,6].

Vancomycin (VNC) and fidaxomicin (FDX) are the two primary agents currently recommended for CDI treatment. Both are effective in achieving initial clinical cure; however, their impact on recurrence rates differs. Emerging evidence suggests that FDX may preserve gut microbial diversity more effectively than VNC. Nonetheless, direct comparative data on their long-term effects on microbiota recovery remain limited [7].

The aim of this study is to characterize the longitudinal evolution of the gut microbiota in patients with CDI. Secondary aims are to compare the impact of different treatment on microbial diversity, and taxonomic composition, and to identify potential microbiome-related factors that may promote sustained clinical cure or predispose to recurrence.

## Results

A total of 143 patients were included: 137 (95.8%) were primary episode, 58 (40.6%) were male, 50 (34.9%) aged >75 years, and 39 (27.3%) immunocompromised. Most episodes were healthcare-associated or hospital-acquired (40.6%), and nearly three quarters of patients required hospitalization. Comorbidity burden was substantial, with a median Charlson Comorbidity Index of 4 (IQR 2–6). Prior healthcare exposure was frequent, including recent hospitalization and antibiotic use in the preceding three months in over 90% of cases. At the time of CDI diagnosis, 44.7% of patients were receiving active antibiotic therapy, and more than half had chronic proton pump inhibitor use. Severe disease or a complicated clinical course was observed in 23.1% of cases, with low rates of sepsis and ICU admission. Recurrence occurred in approximately one fifth of patients (22.4%) (Table 1).Treatment for CDI consisted of VNC in 88 patients (61.5%), FDX in 33 (23.1%), and VNC–BZL in 22 (15.4%). Exposure to non-CDI antibiotics was observed in 52 patients (36.4%) during CDI treatment, while 83 (58.0%) were exposed during follow-up.

Out of the total cohort, 32 (22.4%) patients experienced a first recurrence of *Clostridioides difficile* infection (rCDI), of whom 11 progressed to a second recurrence, and 3 subsequently developed a third episode. The first recurrence was predominantly associated with healthcare-related settings (46.9%), subsequent episodes occurred mainly in community-dwelling patients (p = 0.089). A significant reduction in the requirement for hospitalization was observed as the number of recurrences increased, dropping from 50.0% in the first episode to 18.2% in the second (p = 0.046). Furthermore, therapeutic strategies evolved significantly across episodes (p = 0.024); although FDX monotherapy was the preferred choice for the first recurrence (59.4%), management of the second recurrence shifted toward the use of VNC-based regimens and the intensified use of BZL (36.4%) (S1 Table).

None of the demographic or clinical variables studied were associated with CDI recurrence, which was diagnosed in 32 patients (22.4%) (S2 Table). In a similar vein, CDI treatments were similar between age groups and sex, as was exposure to non-CDI antibiotics during and following CDI treatment. However, immunocompromised patients were prescribed VNC in lower percentages, and received more antimicrobial therapy after CDI (S3 Table). 417 stool samples were

**Table 1.** Demographic characteristics, previous healthcare exposure, comorbidities, clinical course and treatment of *Clostridioides difficile* infection episode, and clinical outcomes.

| | Population [n= 143] |
|---|---|
| **Demographics** | |
| Age, median (IQR), years | 67 (55–79) |
| Age ≥ 75 years old, % (N) | 34.9 (50/143) |
| Males, % (N) | 40.6 (58/143) |
| Community-acquired (ECDC criteria), % (N) | 40.6 (58/143) |
| Healthcare-associated (ECDC criteria), % (N) | 29.4 (42/143) |
| Nosocomial (ECDC criteria), % (N) | 30.1 (43/143) |
| Hospitalization, % (N) | 73.4 (105/143) |
| Previous CDI episode in the past 2 years, % (N) | 4.2 (6/143) |
| **Previous healthcare exposure** | |
| Major surgery in the previous 3 months, % (N) | 14.7 (21/143) |
| Previous institutionalization, % (N) | 3.5 (5/143) |
| Previous hospitalization (last 12 months), % (N) | 55.2 (79/143) |
| Previous antibiotic therapy (last 3 months), % (N) | 91.6 (131/143) |
| Number of previous antibiotic courses | |
| 1, % (N) | 48.9 (64/131) |
| 2, % (N) | 22.1 (29/131) |
| 3, % (N) | 17.6 (23/131) |
| ≥ 4 courses, % (N) | 11.5 (15/131) |
| Active antibiotic therapy at CDI diagnosis, % (N) | 45.0 (59/131) |
| Discontinuation of concomitant antibiotics, % (N) | 27.5 (36/131) |
| Chronic proton pump inhibitor in the previous 3 months, % (N) | 58.7 (84/143) |
| **Comorbidities** | |
| Immunosuppression, % (N) | 27.3 (39/143) |
| Diabetes, % (N) | 22.4 (32/143) |
| Active solid tumor, % (N) | 21.0 (30/143) |
| Moderate or severe renal disease, % (N) | 9.8 (14/143) |
| Congestive heart failure, % (N) | 4.2 (6/143) |
| Chronic respiratory disease, % (N) | 3.5 (5/143) |
| Charlson comorbidity index, median (IQR) | 4.0 (2.0–6.0) |
| **Clinical course and treatment of CDI episode** | |
| Severe colitis | 23.1 (33/143) |
| Sepsis or septic shock associated with CDI | 1.4 (2/143) |
| ICU admission | 0.7 (1/143) |
| First CDI treatment | |
| Vancomycin | 56.6 (81/143) |
| Vancomycin + bezlotuzumab | 14.0 (20/143) |
| Fidaxomicin | 16.1 (23/143) |
| Fidaxomicin extended regimen | 3.5 (5/143) |
| Other fidaxomicin-based regimens | 4.2 (6/143) |
| Other vancomycin-based regimens | 5.6 (8/143) |
| Duration, median (IQR) | 10 (9-10) |
| **Outcomes** | |
| Recurrence, % (N) | 22.4 (32/143) |
| Recurrence time from diagnostic (days), median (IQR) | 26 (19 - 44) |

included: 333 samples from patients with no recurrence (111 at each time point) and 84 samples from the 32 patients with recurrence (Diagnostic (Dx) $n = 32$, End-Treatment (ET) $n = 20$ from patients responding well to the first round of treatment, and recurrence dx (rec), $n = 32$).

## Dynamics of α diversity

All samples included in the diversity analyses met the minimum sequencing depth threshold defined by the rarefaction analysis (10,000 reads per sample, **S1 Fig**). A homogeneity analysis of the Dx samples showed no statistically significant differences in microbial diversity or taxonomic composition according the studied variables.

In the overall cohort of patients, a significant reduction in α diversity was observed after completion of treatment for CDI, as assessed by both the Shannon index (H-index) (Dx: H = 3.14 ± 0.04, ET: H = 2.53 ± 0.06, $p < 0.001$) and the number of ASVs (Dx: 110.20 ± 4.45, ET: 67.39 ± 3.93, $p < 0.001$). In patients whose infection resolved, diversity levels at 8W were significantly higher than those observed at the time of CDI diagnosis ($p < 0.001$; **Fig 1A**). In contrast, in patients who relapsed, the diversity measured in the rec sample remained similar to that in the Dx ($p = 0.69$, **Fig 1B**).

To evaluate the effect of recurrence on microbiota diversity over time using a longitudinal approach, time was included as a continuous variable (defined as the number of days from diagnosis to sample collection). This variable was incorporated into a time-adjusted longitudinal mixed-effects model, yielding results consistent with previous analyses. The linear model showed that patients without recurrence recovered, on average, 0.672 [0.504-0.84], $p < 0.001$) additional ASVs per day, whereas in patients with recurrence this variable did not increase and instead showed a slight decreasing trend (-0.734, [-1.356-(-0.0112)], $p = 0.021$) (**Fig 1C** and **S4 Table**). Similarly, although less markedly, the H-index remained stable over time in patients with recurrence (-0.011 [-0.024-0.002], $p = 0.088$) ((**Fig 1C** and **S5 Table**).

The patients without recurrence were stratified by treatment, sex, immunosuppression and age to assess their influence on diversity dynamics. Among patients treated with VNC, the previously described pattern was confirmed: there was a 57.0% reduction in ASVs ($p < 0.001$) and a 23.3% reduction in the H-index ($p < 0.001$) at the end of treatment compared to baseline (**Fig 2A** and **S6 Table**). At 8W, both H-index demonstrated recovery, reaching levels significantly higher than those at diagnosis ($p = 0.002$, **Fig 2A** and **S6 Table**). Patients treated with VNC-BZL exhibited a similar pattern during treatment, than patients treated with VNC. In contrast, in patients treated with FDX no significant changes in the H-index were observed over the three time points ($p = 0.15$). A reduction in ASVs of 20.8% was observed in ET samples when compared to the diagnostic samples; however, this decrease was not statistically significant ($p = 0.58$). The 8W samples showed a higher number of ASVs than the diagnostic samples ($p = 0.020$, **Fig 2A** and **S6 Table**). In the ET samples, significantly higher H-index values were observed in patients receiving FDX compared to VNC and VNC-BZL ($p < 0.001$). Samples from patients treated with VNZ-BLC showed an H-index comparable to that of patients treated with VNC. This trend was also observed for ASVs ($p = 0.053$, **S6 Table**).

Sex-based analyses revealed that men followed the global pattern of α diversity recovery, while women did not. In women, H-index and ASV values observed at 8W did not significantly exceed those obtained at diagnosis ($p = 0.20$; $p = 0.16$, **Fig 2B** and **S6 Table**). To investigate whether these differences were linked to a sex-specific response to treatment, we analyzed patients treated with VNC by sex. At 8W, the 38 women treated with VNC exhibited significantly fewer observed ASVs (133.42 ± 9.26 vs 170.53 ± 13.86, $p = 0.024$) and a trend toward a lower H-index (H = 3.39 ± 0.09 vs. H = 3.64 ± 0.09, $p = 0.068$) compared to the 28 men receiving the same treatment. These differences were not observed among patients treated with FDX, in whom both men ($n = 9$) and women ($n = 19$) showed comparable α diversity values at 8 weeks (male: H = 3.23 ± 0.84, ASVs = 147.88 ± 31.86, female: H = 3.30 ± .0.17, ASVs = 143 ± 20.51; $p = 0.90$, $p = 0.82$, respectively).

Similarly, immunocompromised patients did not demonstrate an increase in microbial diversity at 8 weeks compared to baseline, as assessed by either the H-index ($p = 0.31$) or observed ASVs ($p = 0.18$, **Fig 2C** and **S6 Table**). When stratifying by treatment and immunosuppression status, no significant differences in α diversity were detected at 8W among the subgroups.

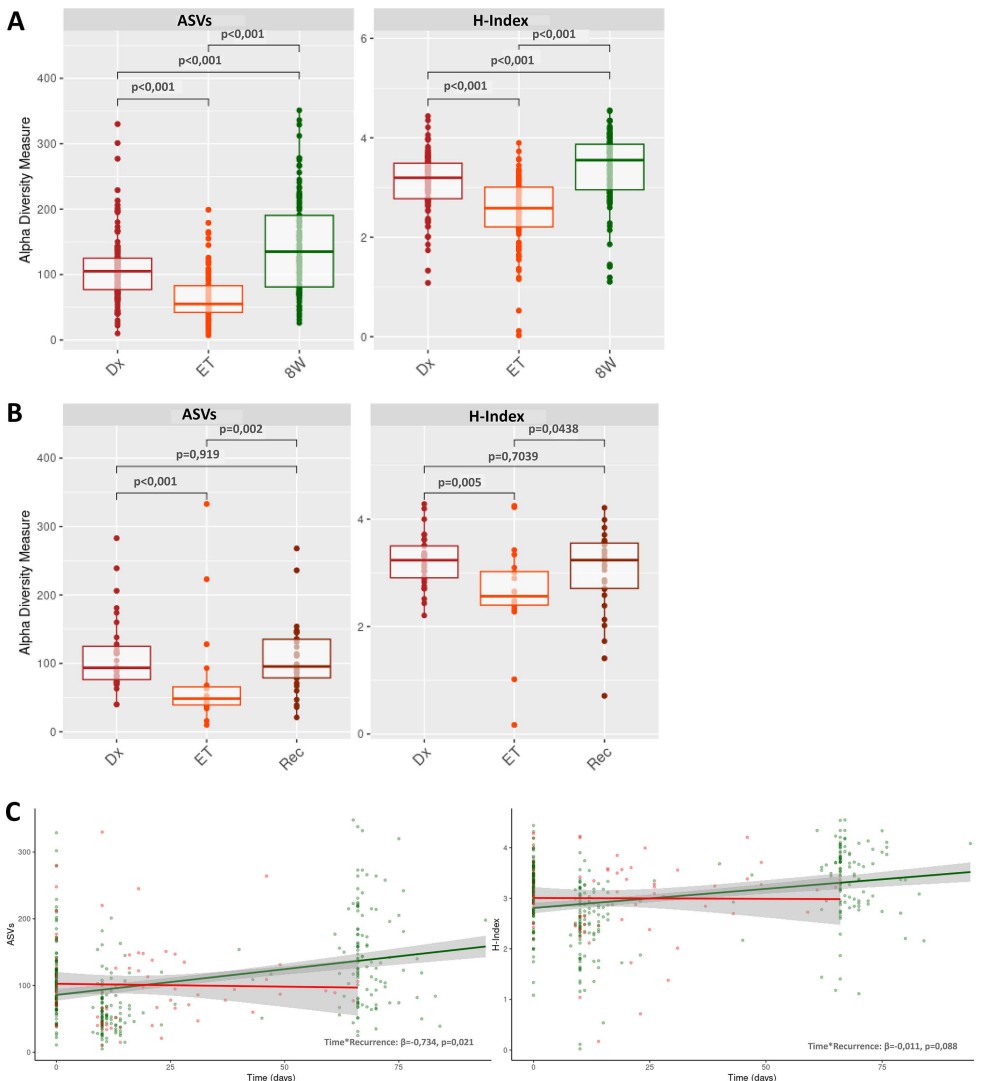

**Fig 1. Evolution of α diversity according to the time points studied.** H-index and ASVs observed in the whole population ($n = 143$) **(A)**, and in recurrent patients ($n = 32$) **(B)**. Results of the longitudinal mixed-effects models generated, in green showed the estimated trend for non-recurrent patients, the red line illustrates the same for recurrent patients **(C)**. Dx = sample from diagnostic, ET: sample from end-treatment, 8W: samples from 8 week post-treatment, rec: sample from recurrence diagnostic.

To evaluate the effect of sex on microbiome recovery minimizing the potential confounding effect of immunosuppression, the same analysis was conducted including only immunocompetent patients. In this subgroup, 39.75% of patients were male, a proportion comparable to that of the overall study population. Consistent with our primary analysis, microbiota diversity recovery at eight weeks differed by sex. In women, diversity at eight weeks (ASVs = 141.73, H = 3.33) remained similar to that observed in the CDI diagnosis sample (ASVs = 113.55, H = 3.18) (ASVs: p = 0.30, H: p = 0.86). However, men exhibited a significantly higher diversity at eight weeks (ASVs = 169.48, H = 3.64) compared with diagnostic sample (ASVs = 112.74, H = 3.13) (ASVs: p = 0.002, H: p = 0.002).No significant differences were observed in any diversity index comparisons based on age category.

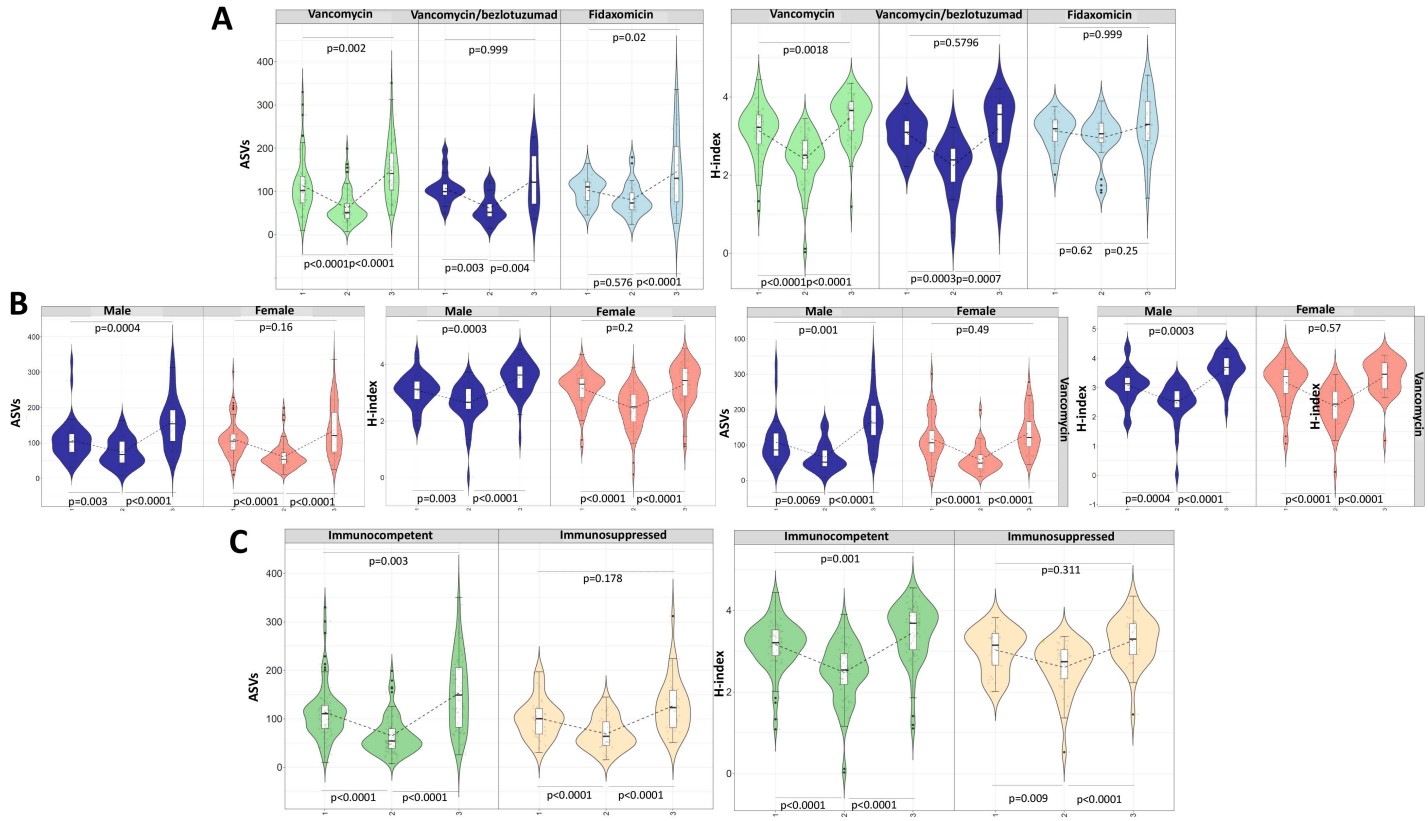

**Fig 2. Evolution of α diversity in non-relapsed patients according to the time points studied (1: diagnosis, 2: end of treatment, 3: 8 weeks post-treatment).** H-index and ASVs observed according to the treatment administered **(A)**, sex **(B)**, and immunosuppression **(C)**.

In order to assess the results globally, all variables that showed a significant association in the univariate analysis (recurrence, CDI treatment, sex, and immunosuppression) were included in a longitudinal linear mixed-effects model. In this model, several subgroups with zero observations were identified, which compromised the robustness of the estimates. Therefore, a final model including recurrence, CDI treatment, and sex was generated. The results of this model were consistent with those previously reported.

For both, observed ASVs and the H-diversity index, a significant effect of time on microbiome recovery was observed depending on recurrence status. Patients with recurrence exhibited a significantly lower recovery of microbiota diversity over time (ASVs: β = −1.936 [−3.147 to −0.725], p = 0.002; H: β = −0.031 [−0.056 to −0.005], p = 0.018). Sex was also associated with differential recovery of observed ASVs over time, with women showing a significantly lower recovery rate (ASVs: β = −0.496 [−0.939 to −0.053], p = 0.028) (**Fig 3A**). With regard to CDI treatment, differences were observed in the longitudinal evolution of the H-index between patients treated with VNC and FDX (β = −0.016 [−0.030 to −0.001], p = 0.032), with the latter group showing less variation in this index over time (**Fig 3B**).

These models were also repeated in the subgroup of patients without recurrence, including immunosuppression as a covariate. The same trends were observed, although statistical significance remained close to the threshold. For ASV richness, the p-value for the interaction between time and sex was 0.077, and for the H-index the p-value for the interaction between time and treatment was 0.078. The results of all fitted models are provided in the supplementary material (**S4 and S5 Tables**).

PLOS Pathogens

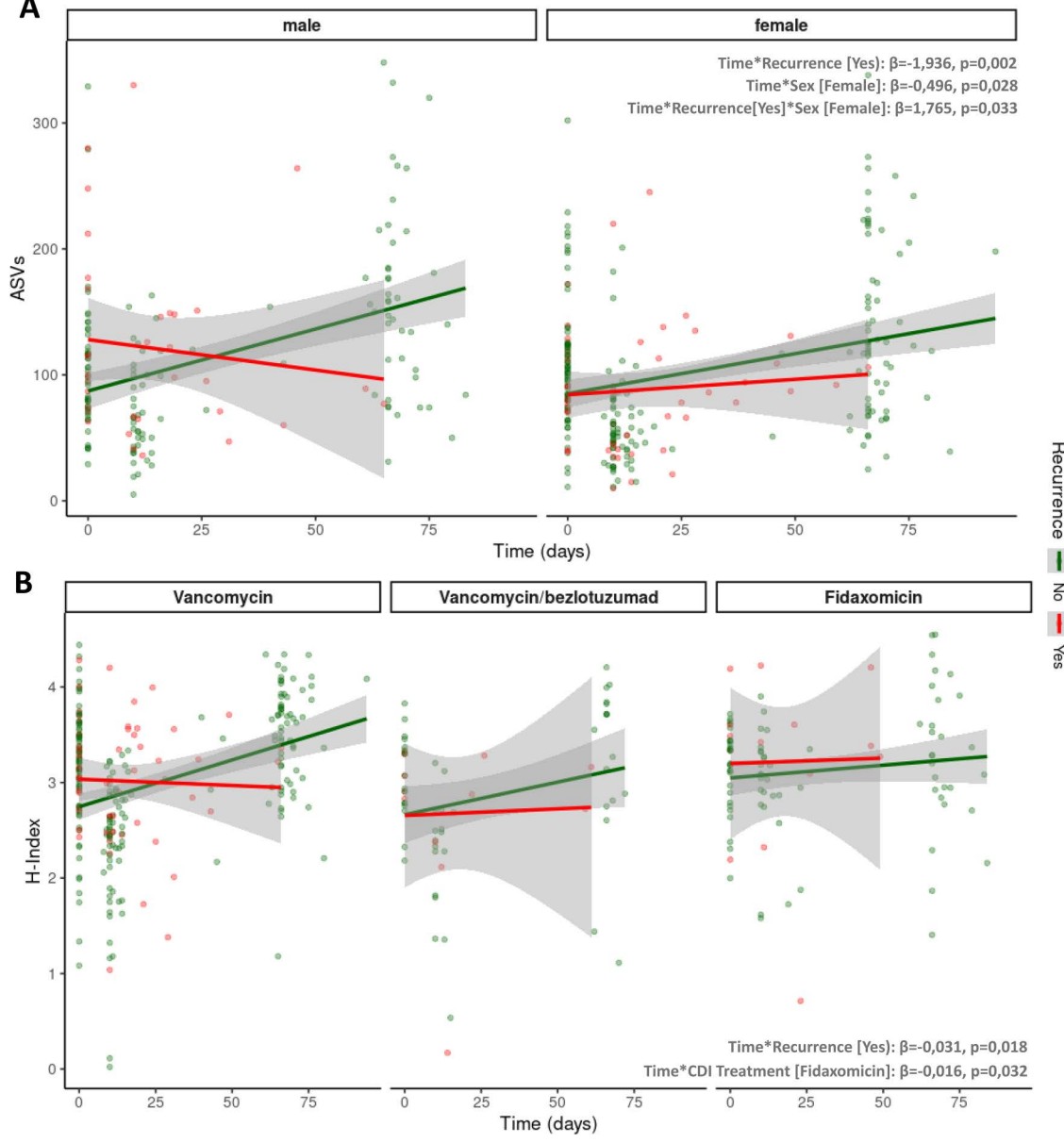

**Fig 3. Longitudinal mixed-effects models of microbial diversity over time. (A)** Observed ASV by sex. **(B)** H-Index by CDI treatment. Points represent individual samples; lines indicate model-predicted trajectories with 95% confidence intervals.

Finally, in patients who experienced recurrence, the previously described treatment- and sex-related trends persisted. Specifically, patients treated with FDX showed higher H-index values after completing treatment compared to VNC and VNC-BZL ($p=0.030$), while women exhibited a significantly lower number of observed ASVs compared to men ($p=0.043$, **S6 Table**).

## Taxonomic composition of the microbiota

**Differences at phylum level.** In the Dx samples of the overall study population, the dominant bacterial phyla were *Firmicutes* (58.1%), *Bacteroidetes* (27.0%), *Proteobacteria* (10.0%), *Verrucomicrobia* (1.6%), *Actinobacteria* (1.5%), and

*Fusobacteria* (1.5%). These relative abundances remained consistent across all subgroups analyzed. No statistically significant differences in phylum-level composition at diagnosis were observed by the studied variables.

In contrast, in the ET samples there were differences according to sex, treatment, and recurrence, but not immuno-suppression or age. Sex-based analysis revealed that women had significantly lower levels of *Proteobacteria* compared to men ($p = 0.018$, **Fig 4A**). Patients treated with FDX showed a higher relative abundance of *Bacteroidetes* than those treated with VNC ($p < 0.001$) or VNC-BZL ($p < 0.001$). Conversely, the proportion of *Proteobacteria* was lower in FDX-treated patients compared to those treated with VNC ($p < 0.001$, **Fig 4B**). Finally, patients with recurrence exhibited significantly lower relative abundances of the phyla *Fusobacteria* ($p = 0.012$) and *Verrucomicrobia* ($p = 0.003$), along with a trend toward reduced *Bacteroidetes* ($p = 0.054$) compared to patients without recurrence (**Fig 4C**).

Related to dynamics of microbiota composition, treatment with FDX did not alter the taxonomic composition of the microbiota, with similar relative abundance percentages maintained for the six phyla studied over the three time points. In patients treated with VNC, the taxonomic composition was modified in five out of the six phyla, and in the case of VNC-BZL, three of these six phyla were altered (**Fig 5A**). In relation to sex, the abundance of *Fusobacteria* did not show

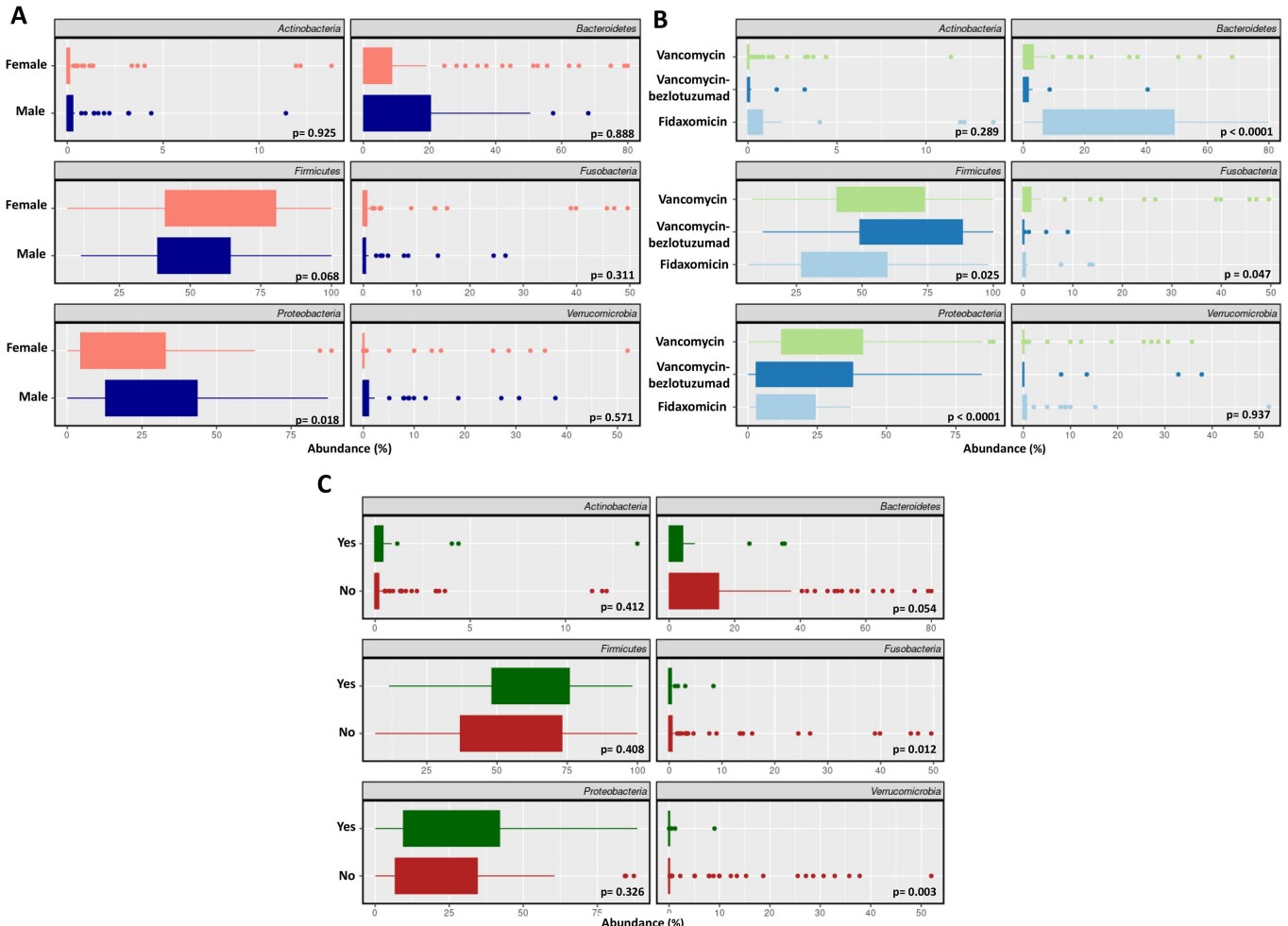

**Fig 4. Phyla with a relative abundance greater than 1% in the ET samples stratified by sex (A), CDI treatment (B) and recurrence (C).** The x-axis represents the percentage of relative abundances.

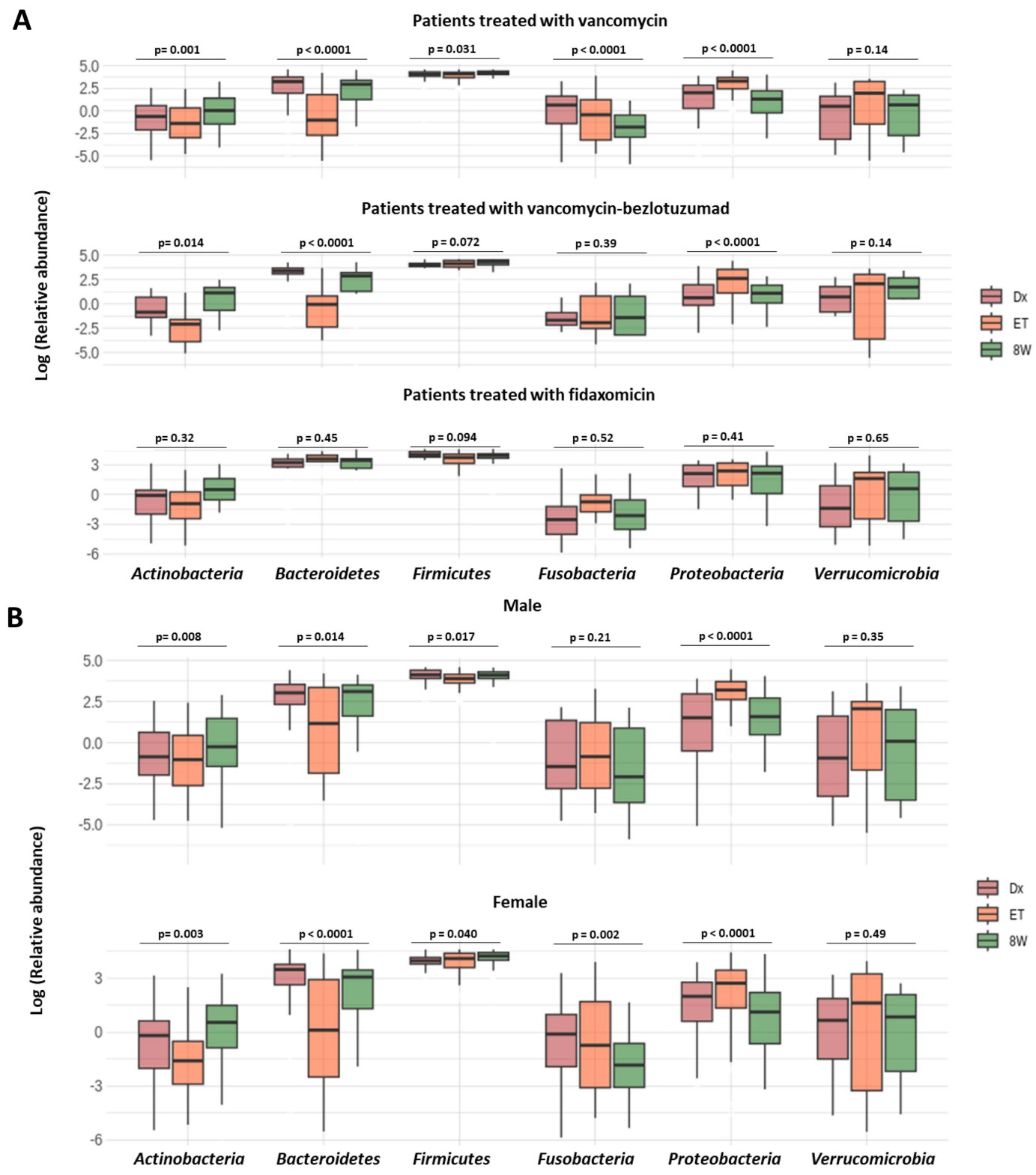

**Fig 5. Evolution of taxonomic composition in non-relapsed patients according to CDI treatment (A) and sex (B).** Dx = sample from diagnostic, ET: sample from end-treatment, 8W: samples from 8 week post-treatment, rec: sample from recurrence diagnostic.

significant changes in men ($p = 0.22$). However, in women it increased then decreased significantly ($p = 0.002$). Similarly, the phylum *Firmicutes* exhibited a similar abundance at diagnosis and 8 weeks post-treatment in men ($p = 0.81$). In contrast, in women, the 8W sample showed a significantly higher percentage of *Firmicutes* compared to the diagnostic sample ($p = 0.034$, **Fig 5B**).

**Differentially represented genera.** No differentially represented genera were identified in the diagnostic samples or at 8 weeks post-treatment when stratified by age, sex, immunosuppression status, or recurrence.

The only time point where the representation of genera showed differences was at the ET. These differences were associated with immunosuppression, treatment, and recurrence, but not sex or age. Immunocompromised patients showed an increased representation of *Anaerostipes* and *Blautia*. Patients who experienced recurrence exhibited a higher relative abundance of the genera *Methanobrevibacter* and *Ruminiclostridium 5*. Patients treated with FDX showed a significantly higher abundance of several genera, including *Bacteroides*, *Faecalibacterium*, *Ruminococcus*, *Lachnoclostridium*, and *Blautia*, as well as members of the family *Lachnospiraceae*, when compared to those treated with VNC. In contrast, no differentially represented genera were observed in patients treated with VNC-BZL versus VNC alone. Finally, patients who experienced recurrence exhibited a higher relative abundance of the genera *Methanobrevibacter* and *Ruminiclostridium 5*.

## Discussion

CDI represents a paradigmatic example of a pathobiont-driven infection—an opportunistic microorganism that exploits alterations in the host microbiome to initiate disease [8]. In a healthy gut, the microbiota is predominantly composed of *Firmicutes* and *Bacteroidetes*, which play a protective role against CDI by competing for nutrients, producing inhibitory substances, and stimulating host immune responses. In contrast, CDI is associated with reduced microbial diversity, including a depletion of beneficial taxa such as *Bacteroidetes, Prevotella,* and *Bifidobacterium* [9,10].

The present study contributes novel data to the understanding of this complex pathophysiological process, examining the longitudinal impact of different CDI treatments on gut microbiota composition and diversity. Related to the overall dynamics of the microbiome, patients with CDI experienced a notable decline in α diversity following completion of antimicrobial therapy. At 8 weeks post-treatment in individuals without recurrence, diversity recovered to levels exceeding those observed at diagnosis. In contrast, patients who experienced recurrence demonstrated similar diversity levels at the time of recurrence as at initial diagnosis, suggesting an impaired capacity for microbiome recovery. This reduced restoration of diversity after treatment-induced depletion may be a factor for increased recurrence risk. To date, few published studies have analyzed the microbiota longitudinally with such a long follow-up and in such a large sample of patients as in our study. Nevertheless, our results are in line with previous reports. In particular, Seekatz et al. [11] showed in a smaller cohort with intra-individual comparisons that patients who experienced recurrence were less likely to recover microbiota diversity, and that recurrence was associated with persistently reduced diversity throughout the CDI episode [12]. These findings support our observation that impaired microbiota recovery is a key determinant of recurrence. Moreover, in our cohort recurrent cases were characterized by reduced microbial richness and diversity, decreased abundance of short-chain fatty-acid-producing bacteria, and an increase in bile salt hydrolase-producing organisms, features associated with impaired microbiome recovery and persistent dysbiosis [13,14].

Failure to restore the gut microbiota to a healthy state in relapsing patients may explain why the use of non-selective drugs that perpetuate dysbiosis, such as VNC, are associated with higher relapse rates, in part due to the loss of secondary bile acid production derived from the microbiome [15]. Thus, several meta-analyses have reported that treatment with VNC results in a higher relapse rate than when using FDX, a drug that is much less damaging to the patient's microbiome [16,17]. In line with this evidence, our work reveals treatment-dependent differences among non-recurrent patients. This discrepancy may be attributed to the broader-spectrum activity of VNC, which disrupts microbiota-derived production of secondary bile acid, compounds essential for maintaining healthy microbiota [18]. Early microbiome responses to CDI

therapy have been shown to correlate strongly with recurrence risk [14,19]. Patients treated with FDX exhibited fewer disturbances in the gut microbiome, and retained a higher relative abundance of health-associated genera such as *Bacteroides*, *Faecalibacterium*, and *Ruminococcus*, when compared with those treated with VNC. These findings support the role of FDX in preserving microbiota integrity, which is crucial for preventing recurrence, and are agree with previous studies showing that VNC causes greater dysbiosis in the microbiome in patients with CDI [20], while decreasing the proportion of some genera associated with a healthy microbiome, such as *Bacteroides* [21]. In this regard, members of the phylum *Bacteroidetes* appear to be critical to maintaining gut homeostasis. For all these reasons, FDX has been consistently identified as a protective agent against CDI recurrence [16,22,23]. Our findings are consistent with prior reports showing that FDX has a limited deleterious effect on this phylum compared to VNC [20,21,24].

Clinical factors also modulated microbiome recovery. We observed that women and immunocompromised individuals exhibited limited recovery of microbial diversity by week 8, with the diversity index remaining comparable to that at diagnosis. Furthermore, VNC treatment was associated with a more pronounced reduction in microbial diversity in women than in men. Sex-based differences were also noted in the behavior of certain phyla, such as *Fusobacteria* and *Firmicutes*, which are key to microbiota stability [25]. These findings may help explain why immunosuppression is recognized as a risk factor for CDI recurrence [26]. In relation to gender, there is a dearth of research addressing the differential effect of CDI treatment in men and women. Our data revealed that VNC damages the microbiota more in women than in men, so it would be interesting to conduct further studies in this area. In fact, some studies are beginning to show female sex is associated with a higher risk of complications and recurrences in the course of CDI [27], and also a higher risk of acquiring this infection [19].

## Limitations

Our work contributes valuable data that help to explain the complex phenomenon of the interaction between the patient's microbiome and the microorganism causing CDI. Patients were enrolled consecutively, but the exclusion of those who did not have samples available for all time points may have introduced a risk of selection bias. Furthermore, microbiota were studied using 16S ribosomal gene amplification, which, although it is the most widely used and cost-effective technique in clinical studies, does not allow in-depth analysis of the relationship between the different microbial communities, nor the presence of genes that indicate their functionality. Likewise, transcriptomics and metabolomics studies were not included. However, the studies performed provide significant data on the influence of drugs on the microbiome and the different interaction of these drugs according to sex.

## Materials and methods

### Ethics statement

This project was conducted with the written approval of the Clinical Research Ethics Committee (CEIC) of the Virgen de Valme University Hospital (reference: 1254-N-20).

### Study design, patients, and setting

The CDI-ANCRAID-SEICV cohort (ClinicalTrials.gov Identifier: NCT04801862) is a prospective, multicenter study conducted across eight hospitals in Spain. Established in 2020 the cohort is coordinated by hospital antimicrobial stewardship teams and systematically includes patients with CDI based on both microbiological confirmation and compatible clinical presentation. Stool samples were obtained at three time points: diagnosis, end-treatment, and 8 weeks post-treatment or at the time of recurrence. All samples are cryopreserved at −80°C for downstream analyses. Patients were included who had provided samples at all three study time points and who had completed the 12-week follow-up before December 31, 2024.

## Variables and data collection

Primary outcome variables was gut microbiota diversity and composition, evaluated using standard α-diversity indices (Shannon index [H] and amplicon variant sequences [ASVs]). Diversity was assessed at three predefined time points, in order to capture the longitudinal evolution of the gut microbiome. Taxonomic composition was also collected, based on the relative abundance of dominant phyla (defined as those representing >1% of total abundance) and differentially represented genera.

Explanatory variables, including patient age (dichotomized as < 75 vs. ≥ 75 years) and sex, were collected for all participants. CDI-related clinical variables comprised the treatment administered (VNC, FDX, VNC-bezlotuzumab (VNC-BLZ) and the presence of recurrence. Recurrence was defined according to the criteria of the IDSA [28]. *Treatment for C. difficile infection was selected in accordance with current ESCMID* [29] and IDSA [30] guidelines, and within a structured, multidisciplinary clinical pathway for systematic case assessment and treatment indication [31]. VNC or FDX was used for initial episodes based on recurrence risk, and FDX or VNC plus BZL for first recurrences. Clinical variables were concomitant antibiotic therapy during CDI treatment and/or follow-up, and immunosuppression.

Study data were collected and managed using REDCap [32,33].

## Microbiota analyses

Microbial DNA was isolated from the stool samples using the QIAamp PowerFecal Pro DNA kit (Qiagen). Microbiota amplicon sequencing was conducted according to the 16S Metagenomics Sequencing Library Preparation protocol (Illumina). V3-V4 regions of the 16S rRNA were amplified by PCR, and sequenced on the MiSeq (600 cycles, 2 × 300 bp).

Raw reads were analyzed using QIIME2 (2021.2) [34]. Denoising was performed with the DADA2. Taxonomy was assigned with SILVA (Release 138) [35]. Prior to diversity analyses, rarefaction curves were generated (**S4 Table**) for the observed ASVs and for the H-index in order to determine the sequencing depth threshold. A minimum sequencing depth of 10,000 reads was defined based on these curves, and only samples that reached this threshold were included in the analyses. H-index and ASVs were calculated, and longitudinal analysis were performed using the R package MicrobiomeStat (1.2). The genera differentially represented were determined using ANCOM-BC (4.1.0), which accounts for the compositional nature of microbiome data and applies an internal correction for multiple comparisons bases on the false discovery rate (FDR). Genera with FDR-adjusted q values < 0.05 and absolute log2 fold ($|log2FC|$) > 2 were considered statistically significant. The robustness of the results was evaluated using pseudcount sensitivity analysis and only accepted the genera when this parameter was true.

## Statistical analysis

The H and ASVs are expressed as means, and the relative abundances and categorical variables as frequencies (percentages). Differences in diversity indices and taxonomic composition among groups were assessed using parametric tests or non-parametric tests, depending on data distribution and variance homogeneity. For paired comparisons across time points, paired t-test or repeated measures ANOVA were used for normally distributed data, while the Wilcoxon signed-rank test or Friedman's test for non-normally distributed data. For comparisons between independent groups with normally distributed data, Welch's t-test was used. P-values were adjusted for multiple testing using the holm method. All statistical tests were two-tailed, and a *p*-value < 0.05 was considered statistically significant. Analyses were performed using IBM SPSS Statistics v25 (Armonk, NY).

## Highlights

- In patients with non-recurrent *Clostridioides difficile* infection (CDI), microbiota α diversity at 8 weeks post-treatment is higher than at diagnosis.

- In patients with recurrent CDI, α diversity on recurrence is similar to that at diagnosis.

- CDI treatment with fidaxomicin is less damaging to the microbiome than vancomycin (with or without bezlotuzumab).

- Recovery of microbiota diversity at 8 weeks is slower in women and immunosuppressed patients.

## Supporting information

**S1 Table. Characteristics and Management of *Clostridioides difficile* Recurrence Episodes.** FDX: Fidaxomicin, VCN: Vancomycin, BZL: Bezlotuzumad.
(DOCX)

**S2 Table. Recurrence rates according to the clinical and demographic variables analyzed.**
(DOCX)

**S3 Table. Percentages according to sex, age and immunosuppression according to concomitant and post-treatment antimicrobial therapy of CDI, and treatment of CDI.**
(DOCX)

**S4 Table. Mean±se values of α diversity according to study time points and explanatory variables and pvalue for comparison among groups.**
(DOCX)

**S5 Table. Longitudinal lineal mixed effects models.** Observed ASVs.
(XLSX)

**S6 Table. Longitudinal lineal mixed effects models.** H-index.
(XLSX)

**S1 Fig. Rarefaction curves of observed ASV (A) and H-index (B).**
(TIF)

## Author contributions

**Conceptualization:** Maria Paz Ventero, Rocio Herrero, María Dolores Valverde-Fredet, Juan Carlos Rodríguez, Nicolas Merchante, Esperanza Merino De Lucas.

**Data curation:** Maria Paz Ventero, Rocio Herrero, Iryna Tyshkovska, María Dolores Valverde-Fredet, Miguel Rodríguez-Fernández, Pilar González-De-La-Aleja, Esperanza Merino De Lucas.

**Formal analysis:** Maria Paz Ventero, Rocio Herrero, María Dolores Valverde-Fredet, Nicolas Merchante, Esperanza Merino De Lucas.

**Funding acquisition:** Maria Paz Ventero, José Manuel Ramos, Nicolas Merchante.

**Investigation:** Maria Paz Ventero, Rocio Herrero, Iryna Tyshkovska, María Dolores Valverde-Fredet, Juan Carlos Rodríguez, Marta Trigo, Nicolas Merchante, Esperanza Merino De Lucas.

**Methodology:** Maria Paz Ventero, Rocio Herrero, Iryna Tyshkovska, Juan Carlos Rodríguez, Miguel Rodríguez-Fernández, Pilar González-De-La-Aleja, Monica Parra.

**Project administration:** Rocio Herrero, Juan Carlos Rodríguez, Nicolas Merchante, Esperanza Merino De Lucas.

**Resources:** Rocio Herrero, Iryna Tyshkovska, María Dolores Valverde-Fredet, Miguel Rodríguez-Fernández, Pilar González-De-La-Aleja, Marta Trigo, Monica Parra, Ana-Isabel Aller, Silvia Otero, Reinaldo Espindola-Gomez, José

Manuel Ramos, Eva M León, Maria García, Miguel Nicolas Navarrete-Lorite, Jara Llenas-García, Ines Portillo, Francisco Jover, Maria Tasias, Juan Jose Caston, Concepción Gil, David Vinuesa-Garcia, Cristina Gomez-Ayerbe, Francisco J. Martínez Marcos, Nicolas Merchante, Esperanza Merino De Lucas.

**Software:** Maria Paz Ventero, Nicolas Merchante, Esperanza Merino De Lucas.

**Supervision:** Juan Carlos Rodríguez, Nicolas Merchante, Esperanza Merino De Lucas.

**Validation:** Nicolas Merchante, Esperanza Merino De Lucas.

**Writing – original draft:** Maria Paz Ventero, Rocio Herrero, Juan Carlos Rodríguez, Nicolas Merchante, Esperanza Merino De Lucas.

**Writing – review & editing:** Maria Paz Ventero, Rocio Herrero, Iryna Tyshkovska, María Dolores Valverde-Fredet, Juan Carlos Rodríguez, Miguel Rodríguez-Fernández, Pilar González-De-La-Aleja, Marta Trigo, Monica Parra, Ana-Isabel Aller, Silvia Otero, Reinaldo Espindola-Gomez, José Manuel Ramos, Eva M León, Maria García, Miguel Nicolas Navarrete-Lorite, Jara Llenas-García, Ines Portillo, Francisco Jover, Maria Tasias, Juan Jose Caston, Concepción Gil, David Vinuesa-Garcia, Cristina Gomez-Ayerbe, Francisco J. Martínez Marcos, Nicolas Merchante, Esperanza Merino De Lucas.

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
