## [Decision Letter · Decision Letter 0]

14 Dec 2025

Dynamics of the microbiota in patients with Clostridioides difficile: recurrence, treatment, sex, and immunosuppression.

PLOS Pathogens

Dear Dr. Rodríguez,

Thank you for submitting your manuscript to PLOS Pathogens. After careful consideration, we feel that it has merit but does not fully meet PLOS Pathogens's publication criteria as it currently stands. Therefore, we invite you to submit a revised version of the manuscript that addresses the points raised during the review process.

We look forward to receiving your revised manuscript.

Kind regards,

William Navarre

Academic Editor

PLOS Pathogens

Anne Jamet

Section Editor

Editor-in-Chief

PLOS Pathogens

orcid.org/0000-0003-2946-9497

Michael Malim

PLOS Pathogens

orcid.org/0000-0002-7699-2064

**Additional Editor Comments:**

Three experts reviewed your submission. The strengths of this study are both the importance of the question and the impressively large cohort that was analyzed. Reviewers indicate that the research findings are not novel and that a major shortcoming of the submitted work is the lack of context provided against the backdrop of many other similar studies. One reviewer has significant concerns about the methods used for analysis (see reviewer 3). All reviewers feel the reporting of the methods and results were inadequate, and that the manuscript itself is in need of substantial editing to improve clarity.

**Journal Requirements:**

At this stage, the following Authors/Authors require contributions: Maria Paz Ventero, Rocio Herrero, Iryna Tyshkovska, María Dolores Valverde-Fredet, Juan Carlos Rodríguez, Miguel Rodríguez-Fernández, Pilar González-De-La-Aleja, Marta Trigo, Monica Parra, Ana-Isabel Aller, Silvia Otero, Reinaldo Espindola-Gomez, José Manuel Ramos, Eva M Leon, Maria García, Miguel Nicolas Navarrete-Lorite, Jara Llenas-García, Ines Portillo, Francisco Jover, Maria Tasias, Juan Jose Caston, Concepción Gil, David Vinuesa-Garcia, Cristina Gomez-Ayerbe, Francisco J. Martínez Marcos, Nicolas Merchante, and Esperanza Merino De Lucas. Please ensure that the full contributions of each author are acknowledged in the "Add/Edit/Remove Authors" section of our submission form.

https://journals.plos.org/plospathogens/s/submission-guidelines#loc-parts-of-a-submission

4) We notice that your supplementary Figures, and Tables are included in the manuscript file. Please remove them and upload them with the file type 'Supporting Information'. Please ensure that each Supporting Information file has a legend listed in the manuscript after the references list.

**Reviewers' Comments:**

Reviewer's Responses to Questions

**Part I - Summary**

Reviewer #1: 1. The main conclusions largely confirm previously published reports that are unfortunately not considered in this manuscript. The authors need to assess their results in the light of these other studies, noting the concurrence of results. There is always value in validation of results with different cohorts.

Specifically, regarding three of the four highlights listed (lines 44-53), the authors might consider the following manuscripts:

i. Okhuysen PC, Ramesh MS, Louie T, Kiknadze N, Torre-Cisneros J, de Oliveira CM, Van Steenkiste C, Stychneuskaya A, Garey KW, Garcia-Diaz J, Li J, Duperchy E, Chang BY, Sukbuntherng J, Montoya JG, Styles L, Clow F, James D, Dubberke ER, Wilcox M. A randomized, double-blind, Phase 3 Safety and Efficacy study of ridinilazole versus vancomycin for treatment of Clostridioides difficile infection: clinical outcomes with microbiome and metabolome correlates of response. Clin Infect Dis. 2024 Jun 14;78(6):1462-1472. doi: 10.1093/cid/ciad792. PMID: 38305378; PMCID: PMC11175683.

ii. Seekatz, A.M., Rao, K., Santhosh, K. et al. Dynamics of the fecal microbiome in patients with recurrent and nonrecurrent Clostridium difficile infection. Genome Med 8, 47 (2016). https://doi.org/10.1186/s13073-016-0298-8

iii. Louie TJ, Cannon K, Byrne B, Emery J, Ward L, Eyben M, Krulicki W. Fidaxomicin preserves the intestinal microbiome during and after treatment of Clostridium difficile infection (CDI) and reduces both toxin reexpression and recurrence of CDI. Clin Infect Dis. 2012 Aug;55 Suppl 2(Suppl 2):S132-42. doi: 10.1093/cid/cis338. PMID: 22752862; PMCID: PMC3388020

2. As noted in the fourth highlight (lines 52-53), the authors determined that recovery of microbiota diversity at 8 weeks is slower in women and immunosuppressed patients. How many of the immunosuppressed individuals were women? It might be more helpful to only consider immunocompetent men and women when evaluating gender influences on microbiome dynamics.

Reviewer #2: This article has focused on studying how the treatment of C. difficile diarrhoea affects the gut microbiota and its relationship with recurrence. Its strengths are that it has a considerable number of patients, the statistical analysis is well done, and significant results are found for the different treatments. The limitations are that the characteristics of the patients have been poorly defined, and this could be a significant handicap.

Reviewer #3: Review of Ventero et al., “Dynamics of the microbiota in patients with Clostridioides difficile: recurrence, treatment, sex, and immunosuppression”

This manuscript investigates microbiota dynamics in patients with Clostridioides difficile infection (CDI), focusing on recurrence, treatment modality, sex, age, and immunosuppression. The study includes 143 patients and is generally well written. The authors report that patients who experience recurrent CDI exhibit lower microbial diversity compared to those who do not relapse, and that fidaxomicin better preserves gut microbiota relative to vancomycin, either alone or combined with bezlotoxumab. These findings are broadly consistent with previously published studies and therefore largely confirmatory.

The manuscript addresses a clinically relevant topic and includes a substantial cohort. However, the analytical approach requires refinement to ensure robust and interpretable results.

**Part II – Major Issues: Key Experiments Required for Acceptance**

Please use this section to detail the key new experiments or modifications of existing experiments that should be absolutely required to validate study conclusions.required to validate study conclusions.required to validate study conclusions.required to validate study conclusions.

Reviewer #1: (No Response)

Reviewer #2: It is necessary to describe the characteristics of the patients; it has not been specified whether the CDI episode is primary or whether they have had any recurrences. Reference is made to antibiotic use during CDI treatment, but it might be interesting to include previous exposure.

There is no explanation of how the treatment is chosen, why some patients receive fidaxomicin and others vancomycin, only that vancomycin is less commonly used in immunocompromised patients.

It would be interesting to describe the changes in recurrences and after treatment of recurrence.

Reviewer #3: • Analytical Concerns:

o There is no indication that p-values were adjusted for multiple testing. Given the number of comparisons, this is essential to reduce false positives.

o The manuscript states that t-tests were used for parametric data but does not specify which type. Welch’s t-test is generally recommended when sample sizes and variances differ between groups.

o Sampling timepoints are not matched between outcome groups. While relapse timing is inherently unpredictable, comparing fixed post-treatment intervals in non-recurrent patients to variable relapse timepoints introduces bias. Time-adjusted mixed-effects models for Shannon diversity and ASV counts would provide a more robust analysis.

o Figure 1 relies on observed ASVs and the Shannon index. Observed ASVs are highly sensitive to sequencing depth, yet normalization procedures and sensitivity analyses (e.g., rarefaction) are not reported. Apparent recovery in Figure 1 may reflect differences in sequencing depth rather than biological change.

o The impact of treatment (vancomycin vs. fidaxomicin vs. vancomycin + bezlotoxumab) on alpha diversity is not adjusted for key confounders and does not account for repeated measures. Mixed-effects models are standard for longitudinal microbiome data and should be employed.

**Part III – Minor Issues: Editorial and Data Presentation Modifications**

Reviewer #1: 1. Line numbering ends towards the end of Results section. There are also no page numbers.

2. A single table providing demographics of the patient cohort, combining the relevant data from Table 1 and S1 would be valuable for ease of understanding by the reader.

3. Much of the microbiome-related data in Table 1 is more readily observed in Figure 2. Perhaps these data could be moved to supplementary data.

4. Since this paper is focused on antibiotic effects, it would be meaningful to better describe the treatment regimens (dose, length of treatment). It would help to have time of recurrence/relapse also presented in the tables.

5. Bezlotoxumab is referred to variously as bezlotuzumab/bezlotuzumad and should be referred to correctly. Some comment should also be included on whether bezlotoxumab treatment made any difference to microbiome outcome.

6. Please define Dx, and rec when introducing for the first time (Lines 121-122).

7. Maintain consistency with regard to use of H-index/Shannon diversity index in text and figure (line 139, Figures 1 and 2).

Reviewer #2: Figure 4, the label of Fusobacteria is wrong

Reviewer #3: • The terms “H Index” and “Shannon Index” appear to be used interchangeably. It should be explicitly clarified that these refer to the same metric, or corrected if they do not.

• Figure legends.

o Abbreviations used in figures should be defined within the legends for reader clarity.

o In Figure 3, the phrase “according to” could be replaced with “stratified by” for greater precision.

o The legend for Figure 3 states that “percentages are shown in logarithmic scale,” but the figure does not appear to use a logarithmic scale. This should be corrected.

• The manuscript appears to apply an arbitrary cutoff for displaying p-values. It would be preferable to either report all p-values or restrict reporting to those meeting a predefined significance threshold.

PLOS authors have the option to publish the peer review history of their article (what does this mean?). If published, this will include your full peer review and any attached files.). If published, this will include your full peer review and any attached files.). If published, this will include your full peer review and any attached files.). If published, this will include your full peer review and any attached files.

...

Reviewer #1: No

Reviewer #2: **Yes:** Rosa del CampoRosa del CampoRosa del CampoRosa del Campo

Reviewer #3: No

**Figure resubmission:**

**Reproducibility:**



---

## [Decision Letter · Decision Letter 1]

5 Mar 2026

Dear Dr. Rodríguez,

We are pleased to inform you that your manuscript 'Dynamics of the microbiota in patients with Clostridioides difficile: recurrence, treatment, sex, and immunosuppression.' has been provisionally accepted for publication in PLOS Pathogens.

Best regards,

William Navarre

Academic Editor

PLOS Pathogens

Anne Jamet

Section Editor

PLOS Pathogens

Sumita Bhaduri-McIntosh

Editor-in-Chief

PLOS Pathogens

orcid.org/0000-0003-2946-9497

Michael Malim

Editor-in-Chief

PLOS Pathogens

orcid.org/0000-0002-7699-2064

Thanks for your attention to the comments and suggestions provided by the reviewers. All three agree the manuscript is substantially improved. Very minor edits were suggested by Reviewer #1.

Reviewer Comments (if any, and for reference):

Reviewer's Responses to Questions

**Part I - Summary**

Reviewer #1: The authors have largely addressed the concerns raised in prior review.

A couple of minor issues remain.

Reviewer #2: The new version is substantially improved, I believe it can be accepted in its current form.

Reviewer #3: I appreciate the authors work in addressing all of this reviewers' comments and it is my opinion that the MS has been significantly improved. While this study confirms results that have been observed in smaller cohorts the large scale and importance of the work merit significance.

**Part II – Major Issues: Key Experiments Required for Acceptance**

Please use this section to detail the key new experiments or modifications of existing experiments that should be absolutely required to validate study conclusions.required to validate study conclusions.required to validate study conclusions.required to validate study conclusions.

Reviewer #1: (No Response)

Reviewer #2: The new version is substantially improved, I believe it can be accepted in its current form.

Reviewer #3: None.

**Part III – Minor Issues: Editorial and Data Presentation Modifications**

Reviewer #1: 1. Line 141: BZL- first introduction of Bezoltoxumb/Bexzotuzumab in text needs to be spelt out before using abbreviation.

2. S1 legend: please correct spelling (currently Bezlotuzumad)

3. Line 228: “VNZ-BLC” needs to be corrected

4. It would be helpful to include the number of stool samples used for analysis in each group in figure legends.

Reviewer #2: The new version is substantially improved, I believe it can be accepted in its current form.

Reviewer #3: None

PLOS authors have the option to publish the peer review history of their article (what does this mean?). If published, this will include your full peer review and any attached files.). If published, this will include your full peer review and any attached files.). If published, this will include your full peer review and any attached files.). If published, this will include your full peer review and any attached files.

...

Reviewer #1: No

Reviewer #2: **Yes:** Rosa del CampoRosa del CampoRosa del CampoRosa del Campo

Reviewer #3: No

---

## [Editor Report · Acceptance letter]

Dear Dr. Rodríguez,

We are delighted to inform you that your manuscript, "Dynamics of the microbiota in patients with Clostridioides difficile: recurrence, treatment, sex, and immunosuppression.," has been formally accepted for publication in PLOS Pathogens.

Best regards,

Sumita Bhaduri-McIntosh

Editor-in-Chief

PLOS Pathogens

orcid.org/0000-0003-2946-9497

Michael Malim

Editor-in-Chief

PLOS Pathogens

orcid.org/0000-0002-7699-2064